# Pericardial Disease in Patients with Cancer: Clinical Insights on Diagnosis and Treatment

**DOI:** 10.3390/cancers16203466

**Published:** 2024-10-12

**Authors:** Laia Lorenzo-Esteller, Raúl Ramos-Polo, Alexandra Pons Riverola, Herminio Morillas, Javier Berdejo, Sonia Pernas, Helena Pomares, Leyre Asiain, Alberto Garay, Evelyn Martínez Pérez, Santiago Jiménez-Marrero, Lidia Alcoberro, Ernest Nadal, Paula Gubern-Prieto, Francisco Gual-Capllonch, Encarna Hidalgo, Cristina Enjuanes, Josep Comin-Colet, Pedro Moliner

**Affiliations:** 1Cardiology Department, Bellvitge University Hospital, L’Hospitalet de Llobregat, 08907 Barcelona, Spain; llorenzo@bellvitgehospital.cat (L.L.-E.); rramosp@bellvitgehospital.cat (R.R.-P.); herminiomorillas@bellvitgehospital.cat (H.M.);; 2Cardio-Oncology Unit, Bellvitge University Hospital—Catalan Institute of Oncology, L’Hospitalet de Llobregat, 08908 Barcelona, Spain; 3Bio-Heart Cardiovascular, Respiratory and Systemic Diseases and Cellular Aging Program, Bellvitge Biomedical Research Institute (IDIBELL), L’Hospitalet de Llobregat, 08908 Barcelona, Spain; 4Medical Oncology Department, Catalan Institute of Oncology, L’Hospitalet de Llobregat, 08908 Barcelona, Spain; spernas@iconcologia.net (S.P.);; 5Clinical Haematology Department, Catalan Institute of Oncology, L’Hospitalet de Llobregat, 08908 Barcelona, Spain; 6Radiation Oncology Department, Catalan Institute of Oncology, L’Hospitalet de Llobregat, 08908 Barcelona, Spain; lasiain@iconcologia.net (L.A.);; 7Preclinical and Experimental Research in Thoracic Tumors (PRETT), Oncobell, Bellvitge Biomedical Research Institute (IDIBELL), L’Hospitalet de Llobregat, 08908 Barcelona, Spain; 8Centro de Investigación Biomédica en Red de Enfermedades Cardiovasculares (CIBERCV), Instituto Salud Carlos III, 28029 Madrid, Spain; 9Cardiology Department, Hospital Universitari Son Espases, 07120 Palma, Spain; 10Department of Clinical Sciences, School of Medicine, Universitat de Barcelona (UB), L’Hospitalet de Llobregat, 08036 Barcelona, Spain

**Keywords:** pericardial disease, cancer, chemotherapy, radiotherapy, pericardiocentesis

## Abstract

**Simple Summary:**

Pericardial disease is a common and severe complication in patients with cancer, often presenting as acute pericarditis, pericardial effusion, or constrictive pericarditis. Causes include direct tumor invasion, metastasis, and cancer treatments like chemotherapy and radiotherapy. Lung cancer is the most frequent etiology, followed by breast cancer and lymphomas. Early detection and multidisciplinary management are crucial. Acute pericarditis requires careful diagnosis and treatment with NSAIDs and colchicine. Pericardial effusion is commonly incidental but can lead to cardiac tamponade, necessitating pericardiocentesis or a pericardial window. Immunotherapy-related effusions typically respond to treatment cessation and steroids. Constrictive pericarditis, although rare, requires prompt diagnosis and may necessitate surgical intervention. Multidisciplinary care and early intervention are vital for improving patient outcomes and quality of life.

**Abstract:**

Pericardial disease is increasingly recognized in cancer patients, including acute pericarditis, pericardial effusion, and constrictive pericarditis, often indicating a poor prognosis. Acute pericarditis arises from direct tumor involvement, cancer therapies, and radiotherapy. Immune checkpoint inhibitor (ICI)-related pericarditis, though rare, entails significant mortality risk. Treatment includes NSAIDs, colchicine, and corticosteroids or anti-IL1 drugs in refractory cases. Pericardial effusion is the most frequent manifestation, primarily caused by lung cancer, followed by breast cancer, lymphoma, leukemia, gastrointestinal tumors, and melanoma. Chemotherapy, immunotherapy, and radiotherapy may also cause fluid accumulation in the pericardial space. Symptomatic relief for pericardial effusion may require pericardiocentesis, prolonged catheter drainage, or a pericardial window. Instillation of intrapericardial cytostatic agents may reduce recurrence. Constrictive pericarditis, though less common, often develops from radiotherapy and requires multimodality imaging for diagnosis, with pericardiectomy as the definitive treatment. Primary pericardial tumors are rare, with metastases being more frequent. Patients with cancer and pericardial disease generally have poor survival, emphasizing the need for early detection. A multidisciplinary approach involving hematologists, oncologists, and cardiologists is crucial to tailoring pericardial disease treatment to a patient’s clinical status, thereby improving the quality of life and prognosis.

## 1. Introduction

Pericardial disease is a common complication in patients with cancer and is usually associated with an unfavorable prognosis. In necropsy series of patients with cancer, pericardial involvement was observed from 2% to 15–30% of the time [1,2,3].

The main forms of pericardial involvement include acute pericarditis, pericardial effusion, and constrictive pericarditis (Figure 1). The most common manifestation of pericardial involvement is pericardial effusion, which is often an incidental finding [4]. Acute pericarditis and constrictive pericarditis are less common presentations of pericardial disease in cancer patients and are often associated with certain cancer treatments like ICIs or radiotherapy. The neoplasms which most frequently produce pericardial involvement are lung cancer (in both sexes) followed by breast cancer, lymphoma, leukemia, gastrointestinal tumors, and melanoma [5].

Secondary pericardial pathology is most prevalent in patients with cancer, being caused by dissemination of the tumor itself (local invasion, metastatic involvement, or lymphatic obstruction) or as a result of oncological treatment toxicity, such as in chemotherapy, radiotherapy, or immunotherapy, although there are also cases of pericardial infection in the context of immunosuppression [4,6]. Finally, primary pericardial tumors are extremely rare, with pericardial mesothelioma being the most common form.

Early detection of pericardial disease in patients with cancer is essential to delineate the best treatment approach, avoid interruptions of antineoplastic treatment, and improve patients’ quality of life. Cardio-onco-hematology units are crucial for providing multidisciplinary care to patients with cancer who develop cardiac complications, particularly those suffering from pericardial issues [7,8,9].

## 2. Acute Pericarditis

Acute pericarditis in cancer patients may be accompanied by malignant pericardial effusion, a combination which is the most frequent form of presentation. To assess the etiology and appropriate treatment, it is crucial to evaluate the location and stage of the disease, as well as the therapeutic strategies and immunological status of the patient.

### 2.1. Epidemiology

About 5–10% of acute pericarditis may be secondary to an underlying oncologic process [10], resulting in a poor prognosis of oncologic disease [11]. However, the real prevalence of pericarditis in cancer patients remains uncertain and probably underdiagnosed.

### 2.2. Etiology

A wide variety of causes can lead to the development of acute pericarditis.

#### 2.2.1. Tumoral

Direct infiltration of neoplastic cells from nearby structures, obstruction of lymphatic drainage at the mediastinal level, or distant metastasis of primary intra- or extrathoracic tumors by lymphatic or hematogenous route are potential causes.

#### 2.2.2. Cancer Therapies

Anthracyclines, cyclophosphamide, bleomycin, and cytarabine are some of the antineoplastic drugs which can produce pericarditis [12]. The risk of long-term pericardial complications following cancer therapy-induced pericarditis is unknown but is thought to be generally low [13].

Although chronic use of dasatinib (tyrosine kinase inhibitor) may promote the development of pericarditis, it is more frequently associated with pericardial effusion. A potential association for pericarditis and pericardial effusion has also been described with ivosidenib and enasidenib, which are used for the treatment of relapsed or refractory acute myeloid leukemia (AML), and venetoclax, which is used for AML and chronic lymphocytic leukemia [14,15]. Table 1 shows the more frequent cancer therapies which cause pericardial disease.

#### 2.2.3. Radiotherapy

Pericardial disease in patients who had undergone radiotherapy was one of the most frequent forms of cardiac involvement some decades ago [13,16]. Generally, it appears within days to weeks after exposure to radiotherapy, especially in patients with esophageal cancer or Hodgkin’s lymphoma [17]. Late forms of pericardial disease have been described in up to 20% of patients after 15–20 years of treatment with radiotherapy (RT), even without prior acute pericarditis [18].

It is suggested that pericardial involvement after radiotherapy treatment is secondary to the generation of reactive oxygen species which generate pores in the pericardium, favoring neutrophilic infiltration and exudate [16].

The risk of pericarditis increases from 5% to more than 50% if the total dose is increased from 40 to 50 Gy, with the risk also being higher if more than 30% of the heart receives 50 Gy of radiation from the established field [4,17,19]. Typically, radiation doses at the cardiac level are V5Gy < 55 Gy; V10Gy < 33%; V20Gy < 10%; and V40Gy < 5% with a mean dose <4 Gy [20,21]. However, its frequency has currently been considerably reduced due to the use of lower doses, techniques which allow better delimitation of the radiation field (such as intensity-modulated or arc-modulated treatment), and radiation to the left breast during forced inspiration [22].

#### 2.2.4. Immunotherapy

The prevalence of pericarditis in patients treated with immunotherapy is roughly 0.3% [23]. The mean time of onset is between 30 and 70 days [24], although it has been described up to 6 months after immunotherapy initiation. This correlates with a worse prognosis, since a 1.5 fold increase in mortality has been observed compared with patients who do not develop pericarditis [23].

The pathophysiological mechanism is not clearly understood. A greater affinity of T cells for antigens shared by the tumor and cardiac tissue ultimately causing pericarditis has been suggested [4]. In general, acute pericarditis is reversible in about 75% of patients [25], and consequently, the risk of long-term pericardial complications is low [13].

### 2.3. Diagnosis

The diagnostic criteria are identical to those used in the diagnosis of acute pericarditis of any other cause: ECG with diffuse and concave ST-segment elevation, a decreased PR interval, rubbing murmur, typical clinical features, and the presence of pericardial effusion. When acute pericarditis is suspected, concomitant myocardial involvement should be evaluated with the assessment of myocardial biomarkers in order to rule out myopericarditis [18].

The use of imaging tests such as computed tomography (CT) or cardiac magnetic resonance imaging (cMRI) is recommended when diagnostic uncertainties remain after clinical evaluation and echocardiography. The latter, cMRI, is a good choice, given the absence of irradiation, the possibility of characterizing the myocardium and direct study of the pericardium. The presence of late gadolinium enhancement in fat suppression sequences, together with pericardial edema in T2 sequences, indicates active inflammation [26]. Positron emission tomography-CT (PET-CT) is an alternative in which a “ring of fire” surrounding the heart can be observed, reflecting the increased metabolism indicative of acute pericarditis [4,27].

### 2.4. Treatment

Limited data are available to guide the management of acute pericarditis in cancer patients. Therefore, most treatment recommendations are extrapolated from observational studies, routine care in patients without cancer, and expert opinions. The use of non-steroidal anti-inflammatory drugs (NSAIDs) with tapering every 2–4 weeks after the resolution of symptoms, together with the use of colchicine for 3 months, is recommended to reduce the risk of recurrence. NSAIDs should be used with caution in these patients because of the increased risk of bleeding and renal complications [28]. Systemic corticosteroids at low-to-intermediate doses (0.2–0.5 mg/kg/day) have only been indicated in refractory cases, as long as the presence of a concomitant active infection has been ruled out [13,18,26]. In cases of acute pericarditis refractory to conventional treatment, interleukin-1 receptor antagonists such as anakinra and rilonacept have been reported as potentially beneficial [29,30].

Pericarditis resulting from oncospecific treatment may require discontinuation of the therapy, following the recommendations of the multidisciplinary team who is managing the case. Usually, cancer treatment can be continued in cases of uncomplicated pericarditis secondary to immunotherapy, involving NSAIDs or colchicine. Discontinuation of immunotherapy and administration of high-dose corticosteroids (1 mg/kg/day MPDN) and colchicine should be considered if pericarditis is severe and accompanied by moderate-to-severe pericardial effusion [13,18,25]. Resumption of immunotherapy after the resolution of pericardial disease should be discussed within the multidisciplinary team, and a close follow-up is warranted [13].

## 3. Pericardial Effusion

### 3.1. Epidemiology

Pericardial effusion is a relatively common incidental finding in cancer patients and entails poor prognosis and an advanced stage of the disease [2,26,31,32]. Malignant pericardial effusions are characterized by a recurrent course, large amounts of fluid (Appendix A), presentation of cardiac tamponade, the absence of response to empirical treatment with NSAIDs, the absence of acute phase reactants, and hematic fluid [33,34].

Cancer treatment is responsible for about 30% of cases of pericardial effusion, although the percentage could be higher with the increased use of immunotherapy. Neoplastic origin accounts for more than 30% of all pericardial tamponades [35], and it is estimated that 30–50% of malignant pericardial effusions progress to pericardial tamponade [5,34].

Overall, it should be noted that in up to two thirds of patients with pericardial effusion and neoplasia, pericardial effusion is not attributed to malignancy but inflammation, infection, or toxicity. However, these data rely on the sensitivity of the cytology of drained pericardial fluid, ranging from 67% to 92% [6,18,26,33]. Therefore, malignant cytology of pericardial fluid should be used mainly as a diagnostic confirmation tool. In cases with a high degree of suspicion of malignant etiology, a negative cytology may not exclude cancerous etiology.

### 3.2. Etiology

The development of pericardial effusion in patients with cancer can be due to different etiologies, and thus a comprehensive diagnostic work-up is recommended.

#### 3.2.1. Tumoral

Lung cancer is the most common malignancy affecting the pericardium, followed by breast cancer, upper gastrointestinal, pancreas, melanoma, and hematologic neoplasms (especially B cell lymphomas). Tumor invasion of the mediastinal lymph nodes may also obstruct the lymphatic drainage of the pericardium, causing accumulation of pericardial fluid [31]. Overall, neoplastic cells may reach the pericardium by direct invasion or lymphatic or hematologic dissemination.

#### 3.2.2. Cancer Therapies

The main drugs which can cause pericardial effusion are anthracyclines, cyclophosphamide [36], cytarabine [37,38], and specific tyrosine kinase inhibitors like dasatinib (with a 29% described incidence) [26,39,40,41,42]. The underlying mechanisms range from the generation of oxygen-free radicals and oxidative stress to increased endothelial permeability, as in the case of dasatinib [26].

#### 3.2.3. Radiotherapy

It can cause acute or late pericardial effusion months or decades later, usually in 10% of patients. In some series, up to 50% incidence has been described, especially in non-small cell lung cancer [43].

The fluid generated is usually fibrin-rich or hemorrhagic. It is thought to be the result of microvascular damage produced by ionizing radiation, which alters venous and lymphatic drainage of the pericardium, together with the inflammation induced by the radiation itself [33]. Nevertheless, current radiotherapy techniques seek to minimize collateral irradiation of the heart. It should be noted that the risk of pericardial effusion after RT increases in patients who have received previous cardiotoxic treatments (anthracyclines, platinum-derived agents, taxanes, gemcitabine, bevacizumab, and immune checkpoint inhibitors) [31].

#### 3.2.4. Immunotherapy

Immune checkpoint inhibitors (ICIs) have been associated with toxicities called immune-mediated adverse events (imAEs). Pericardial toxicity is the second most frequent type with a prevalence of about 7–14% [44,45], especially secondary to treatment with nivolumab and, less frequently, ipilimumab [46,47,48]. Pericardial involvement secondary to ICIs occurs early, with a mean time of 30 days from the first dose and being severe in 80% of cases, with a mortality rate of 21% [43].

The pathophysiological mechanism is uncertain. It has been observed that RT may expose pericardial antigens which are subsequently recognized by T lymphocytes activated by immunotherapy [49]. Other authors defend the concept of “pseudoprogression” due to the invasion of T lymphocytes [50] or the presence of micrometastases in the pericardial fluid. Cytology is positive for malignant cells in about 50% of pericardial effusions in patients under treatment with ICIs [51].

#### 3.2.5. Other

Heart failure, impaired liver function, hypoalbuminemia, and chronic renal failure can also cause pericardial effusion. Other possible etiologies are the presence of pneumonia or empyema causing purulent pericardial effusion, coexisting connective tissue diseases, thoracic interventions and opportunistic infections by cytomegalovirus, tuberculosis, and fungi such as *Candida* or *Aspergillus* [31,32]. Chylopericardium is a rare cause of pericardial effusion, usually accompanied by chylothorax, which may be encountered after iatrogenic thoracic duct lesion during thoracic surgeries or in the context of mediastinal tumors like lymphoma [52].

### 3.3. Diagnosis

The clinical course depends on the magnitude of the effusion and the rate of progression, although most are asymptomatic [31,33]. Malignant pericardial effusions are usually moderate or severe. If a rapidly developing pericardial effusion occurs, then even a little pericardial effusion may increase the pressure in the pericardial space, compress the heart chambers, and precipitate a cardiac tamponade [33,53].

Transthoracic echocardiography remains the gold standard in pericardial effusion assessment, since it allows direct observation of the presence and quantity of pericardial fluid, and at the same time, it enables evaluating the signs of cardiac tamponade [18]. Cardiac CT and MRI have a complementary role in characterizing pericardial effusion, especially in evaluation of the presence of hematic effusion, pericardial thickening, calcifications, or suspicion of effusive-constrictive pericarditis [5].

The definitive diagnosis of malignant pericardial effusion can only be made after analysis of the pericardial fluid [4,5]. In cases without hemodynamic repercussions, a cytological study of pericardial fluid can be performed by pericardiocentesis [18] in order to detect abnormal malignant cells. Cytology is positive in about 50% of malignant pericardial effusions [5,32,35] and has been associated with lower survival in patients with known malignancy [32,33,54]. A negative cytology may suggest alternative etiologies of the effusion, such as cancer treatment, lymphatic obstruction, or opportunistic infections [32].

Pericardial biopsy via thoracoscopy or thoracotomy may be performed when the etiology of pericardial effusion persists uncertainly after a comprehensive diagnostic work-up. Nevertheless, it is an invasive procedure, and it is complex to obtain a representative specimen [13].

The use of markers in pericardial fluid such as carcinoembryonic antigen (CEA), neuron-specific enolase (NSE), carbohydrate antigen 19-9 (CA 19-9), carbohydrate antigen 72-4 (CA 72-4), and squamous cell carcinoma (SCC) is controversial. In spite of higher levels being observed in malignant pericardial effusions, none of them have demonstrated sufficient accuracy to distinguish between benign or malignant origins of effusions [18,55]. CA 72-4 has shown the highest diagnostic accuracy in favor of the neoplastic origin of effusions [55].

### 3.4. Treatment

The aim of treatment is to improve quality of life, alleviate symptoms, and prevent recurrences, as well as maintain clinical conditions which allow the continuation of cancer treatment.

The first-line treatment for pericardial effusion is pericardiocentesis (Figure 2). This procedure is indicated in moderate or severe symptomatic effusions, hemodynamic instability, pericardial tamponade, absence of response to treatment, or when a bacterial or fungal etiology is suspected. In addition, it is recommended to add extended catheter drainage for 2–5 days to promote adherence of the pericardial layers until there is no more or minimal effusion (less than 30 mL within 24 h). This procedure does not require general anesthesia. After puncturing and drainage, a catheter (extended drainage) may be placed to facilitate complete evacuation of the fluid. The most common access approach is the subxiphoid approach followed by apical access. Less frequently, left parasternal access can be used. The most frequent adverse effects of this technique include supraventricular arrhythmias due to catheter irritation, which is usually asymptomatic [33], and infection in cases where the catheter is maintained for more than 7 days [31,56]. Mild-to-moderate pericardial effusions (4–20 mm) can be conservatively managed with close follow-ups and monitoring, initially with reevaluation 7–14 days after initial diagnosis and subsequently every 4–6 weeks [13].

In the case of pericardial effusions secondary to ICIs, drainage is not usually required since they normally respond to treatment interruption and corticosteroid therapy [13,51,57].

A pericardial window can be performed in those cases in which percutaneous access is not feasible (due to a great distance between the xiphoid appendix and the pericardial cavity, hepatomegaly, or solid tumors with adherence to the pericardium) or severe effusion with rapid growth. This technique allows drainage of the liquid to the pleural cavity with a greater absorptive capacity [13,32]. This can be performed by subxiphoid access or by video thoracoscopy. The recurrence rate in both cases is about 5.7%, and the risk of complications is about 4.5% [58,59]. A pericardial window may not be urgently available, and it requires a longer postoperative recovery time compared with pericardiocentesis, which may delay administration of the planned cancer treatment [32].

Cancer patients are often on anticoagulation therapy due to cancer-associated thrombosis or atrial fibrillation [60]. The coexistence of pericardial effusion and the need for anticoagulation therapy results in a challenging clinical scenario. There are no specific recommendations for the management of anticoagulant therapy in pericardial effusion in patients with cancer. In clinical practice, the interruption of anticoagulation therapy and its resumption depend on the progression of the pericardial effusion, the planned treatment (colchicine, percutaneous or surgical treatment, etc.), and the indication of anticoagulation [18].

### 3.5. Recurrence Prevention

The recurrence rate after pericardiocentesis is roughly 38%, with a reduction in recurrence to 12.1% if associated with extended catheter drainage, 10.8% if intrapericardial cytostatics are administered, and 10.3% if combined with balloon pericardiotomy.

According to a systemic review by Virk et al. [61], the periprocedural mortality rate of pericardiocentesis is less than 1%. Intrapericardial administration of cytostatic agents is one of the strategies used to reduce the number of recurrences since they promote adhesion between the two pericardial layers, reducing the risk of fluid accumulation [33]. The main substances used are cisplatin (9.1% recurrence and 13% complications), tetracycline (8.8% recurrence but high rate of complications (45%)), thiotepa (9.3% recurrence and 6.7% complications), bleomycin (11.1% recurrences), and finally mitomycin C and mitoxantrone, with little use [59]. Bleomycin showed superiority in terms of a lower percentage of adverse effects and hospital stays compared with doxycycline. Its effectiveness, associated with pericardiocentesis, has also been recently defined, with a low rates of recurrence (3.2%) and complications [62], although there are no comparative studies evaluating bleomycin with other drugs [63]. However, a greater effectiveness of intrapericardial cisplatin has been described in patients with lung cancer and pericardial disease compared with systemic chemotherapy, with the best results being obtained with a combination of local (intrapericardial) and systemic chemotherapy [3]. In the case of breast cancer, a potential benefit of thiotepa over other substances has been described [18]. The side effects of sclerosing agents include chest pain (which usually responds to conventional analgesia), fever, skin toxicity, arrhythmias, and increased risk of constrictive pericarditis [58,61]. However, intrapericardial cytostatics are essentially used in patients with a low life expectancy, and thus the latter complication is not usually observed [62].

Balloon pericardiotomy requires a percutaneous subxiphoidal pericardiocentesis followed by various balloon inflations through the parietal pericardium. This allows direct communication between the pericardial cavity and the peritoneum or the pleura with drainage (90–97%) in cases of highly extensive pericardial effusion or recurrent cardiac tamponade [6,13]. Approximately 30% of complications have been described, with the most frequent ones being chest pain, fever, pneumothorax, bleeding of epicardial vessels, and pleural effusion requiring drainage. Even though some authors advocate for this technique as the first choice due to its effectiveness and safety [64], more studies are needed to evaluate its role in this clinical context [59,61].

The surgical approach is unusual in patients with cancer. Surgical pericardiotomy showed no difference in symptom improvement compared to pericardiocentesis; the recurrence rate is similar, and it is associated with a higher rate of complications [18,32]. Pericardiectomy is rarely indicated, except in cases of constriction in patients with good life expectancy or complications of previous procedures [18].

Recently, a single-center observational retrospective study explored the usefulness of the administration of 0.6 mg of colchicine twice a day for two months in 445 patients who underwent pericardiocentesis for malignant pericardial effusion. A statistically significant reduction in all-cause mortality and a need for new pericardiocentesis or pericardial window were observed. Moreover, it was found that after removal of pericardial drainage, 70.7% had pericardial adhesions, and 36.5% had constriction during follow-up [65]. Currently, there is no clear consensus on the use of nonsteroidal anti-inflammatory drugs and colchicine after the procedure, but colchicine could be considered because it is safe and tolerated well.

There are few non-randomized studies comparing the different types of percutaneous and surgical treatment. The optimal treatment of these patients remains controversial, depends on the availability and experience of the different procedures in each hospital, and requires a multidisciplinary approach. The stage of the disease, expected response to oncological treatment, and patient’s frailty and life expectancy are also crucial for selecting treatments in order to improve quality of life, alleviate symptoms, prevent recurrences, and enable oncological treatment [5].

## 4. Constrictive Pericarditis

### 4.1. Epidemiology

Constrictive pericarditis is characterized by pericardial stiffness which hinders the correct filling of the cardiac cavities, generating an increase in intracardiac pressures and leading to clinical signs of congestion or low cardiac output [58]. The incidence of constrictive pericarditis in patients with cancer is not well established and is probably underestimated. In this setting, effusive-constrictive or constrictive pericarditis usually occurs (4–20%) and is frequently associated with the appearance of pericardial effusion in the subacute phase [18].

### 4.2. Etiology

About 54% of constrictive pericarditis cases are idiopathic, 17% are a consequence of acute pericarditis, 11% are secondary to tuberculosis, 7% are purulent pericarditis, and in fewer than 10% of cases, the etiology is previous cardiac surgery or radiotherapy [66]. The risk of progression to constrictive pericarditis depends on the etiology, being low (<1%) in viral and idiopathic pericarditis, intermediate (2–5%) in immune-mediated and neoplastic pericarditis, and high (20–30%) in bacterial pericarditis, especially for purulent cases [18,67].

### 4.3. Diagnosis

Transthoracic cardiac ultrasound plays a key role in constrictive pericarditis diagnosis. Three characteristic parameters of this pathology have been described: ventricular septal shift, velocity e’ of the medial mitral annulus ≥ 9 cm/s (or annulus reversus), and diastolic flow reversal in suprahepatic veins. The combination of ventricular shift with one of the other two characteristics presents a sensitivity of 87% and a specificity of 91% [68].

Cardiac MRI is also useful in identifying constrictive physiology, pericardial thickening, and active inflammation in cases with transient constriction. Cardiac CT allows identification of pericardial calcifications and planning of surgery, although it is not essential for diagnosis. Most cases in the chronic phase present pericardial thickening or calcifications, although in 20% of cases, there may be constriction with a normal pericardial thickness [18]. A definitive diagnosis is obtained by performing a right heart catheterization. However, this procedure is reserved for those cases in which diagnostic uncertainty persists despite noninvasive tests [4,69].

### 4.4. Clinical Features and Treatment

Patients with constrictive pericarditis usually present with symptoms of heart failure or low cardiac output in the absence of previous cardiomyopathy. Physical examination reveals Kussmaul’s sign, a paradoxical pulse, pericardial rubbing murmur, jugular ingurgitation, and peripheral edema [4,69].

There are three constrictive pericardial syndromes [18]: transient constriction (spontaneous resolution or after anti-inflammatory treatment), effusive-constrictive pericarditis (when pericardial effusion coexists with signs of constriction), and chronic constrictive pericarditis (when constriction persists beyond 3–6 months). In the cardio-onco-hematology setting, the most frequent scenarios are the effusive-constrictive phase after pericardial effusion recurrences or extensive pericardial infiltration (both situations with poor prognosis) and late constrictive pericarditis after thoracic radiotherapy, which may have worse evolution than idiopathic forms due to myocardial involvement [70].

The treatment of constrictive pericarditis consists of pericardiectomy, with a described perioperative mortality rate of 11% [3]. However, the surgical risk in patients in the oncological setting must be balanced with the potential therapeutic benefit.

Empirical treatment with anti-inflammatory drugs for 2–3 months may be considered in cases with transient constriction or newly diagnosed if there is evidence of pericardial inflammation (elevated CRP or enhancement in cardiac MRI) [18,69]. Anakinra might be potentially beneficial for the treatment of constriction in patients with recurrent or incessant pericarditis, as it is corticodependent and refractory to colchicine [71].

## 5. Pericardial Tumors

Pericardial tumors are far less frequent than the rest of oncologic pericardial pathology. Most tumor masses correspond to metastases (20–40 times greater than primary tumors) which reach the pericardium through hematic or lymphatic dissemination [33], as 12% of patients with cancer present pericardial metastases at autopsy. Melanoma is the tumor with the greatest avidity for the pericardium, although the most frequently detected tumors are thoracic (lung, breast, and esophagus) [72].

Primary pericardial tumors are infrequent among cardiac tumors (0.001–0.007%), with most of them being benign [17]. They are usually located in the cardiophrenic angle, especially on the right side, and include lipomas, hemangiomas, teratomas, fibromas, and pericardial cysts. In contrast, among malignant pericardial tumors, mesothelioma is most prevalent, accounting for 50% of all primary pericardial tumors, and it is characterized by the formation of multiple pericardial masses [4,33]. Dissemination to other structures is infrequent and is associated with poor 1 year survival [73]. As for benign pericardial tumors, surgical resection is usually indicated if they are symptomatic or generate a hemodynamic compromise. However, surgery seems to have a limited impact on survival, and patients receiving platinum-based chemotherapy achieve longer survival outcomes [74].

## 6. Long-Term Follow-Up and Prognosis

Clinical guidelines recommend performing a follow-up echocardiography every 5 years for patients with a history of acute pericarditis during radiotherapy with a radiation field which includes the heart and who are at high risk of developing constrictive pericarditis [13]. Patients with cancer who present pericardial involvement often have a poor prognosis [18,32,58], although this depends on the clinical presentation, the underlying disease, and the existence of a targeted cancer therapy. Specifically, shorter survival has been observed when the underlying neoplasm is lung cancer compared with breast cancer [33,56], although immunotherapy has displayed promising results in the prognosis of these patients.

## 7. Future Directions

Future advances in the treatment of pericardial disease will focus on early detection and targeted therapy to improve the quality of life and prognoses of patients with cancer. Personalized medicine, using genetic and molecular profiling, aims to develop targeted therapies with fewer cardiotoxic effects. Understanding the impact of immunotherapy on the pericardium, particularly immune checkpoint inhibitors, chimeric antigen receptor (CAR-T) cell therapies, and other emerging therapies, is critical. An evidence-based approach to medicine is required. Conducting robust clinical trials and establishing multicenter registries will address knowledge gaps and guide future research. Emphasizing multidisciplinary approaches involving oncologists, hematologists, and cardiologists will improve patient care through shared protocols. In this regard, the creation of cardio-oncology-hematology units is essential to easing communication between medical professionals, organizing the care process, and thus offering the best possible treatment to cancer patients.

## 8. Conclusions

Pericardial pathology is frequent among patients with cancer, and it is often present at the time of tumor diagnosis. A comprehensive diagnostic approach is crucial since early treatment may allow prompt resumption of cancer therapies. Pericardial effusion is the most common clinical syndrome, but there is limited evidence on its management. Complementary measures to pericardiocentesis such as colchicine treatment, sclerosing agents, and balloon pericardiotomy can reduce recurrences. Cardio-oncology units facilitate individualized decisions based on a patient’s clinical situation and following the recommendations of the multidisciplinary team approach.

## Figures and Tables

**Figure 1 cancers-16-03466-f001:**
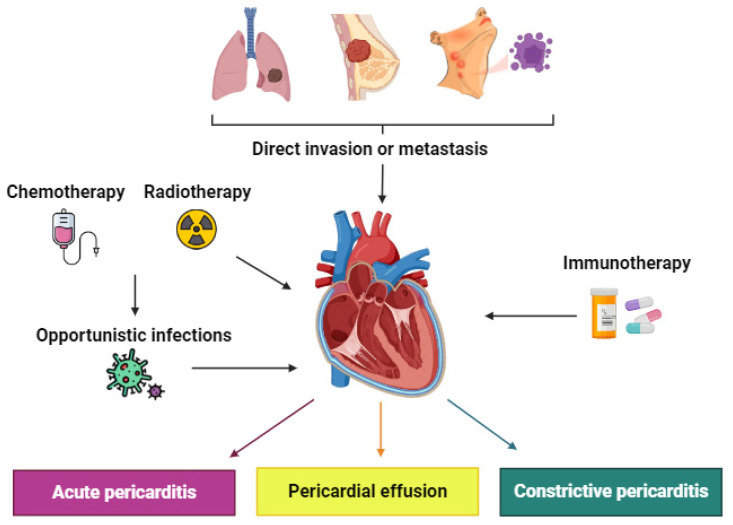
Clinical manifestations of pericardial disease.

**Figure 2 cancers-16-03466-f002:**
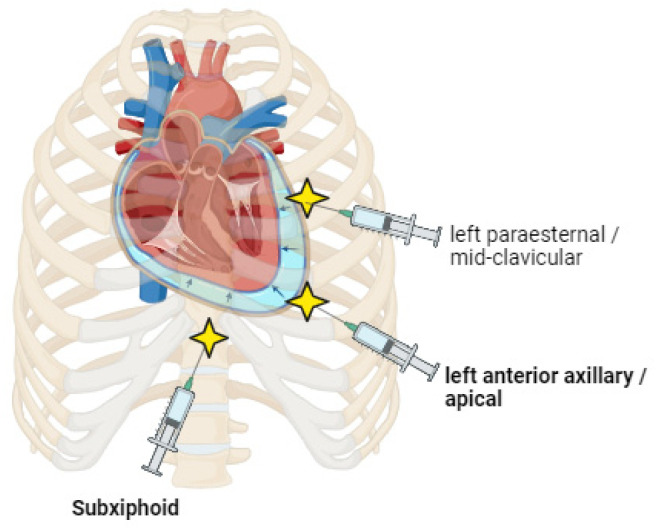
Pericardiocentesis approaches (subxiphoid, apical, and left parasternal).

**Table 1 cancers-16-03466-t001:** Cancer therapies causing pericardial disease.

Acute Pericarditis	Pericardial Effusion
Anthracyclines	Anthracyclines
Cyclophosphamide	Cyclophosphamide
Bleomycin	Cytarabine
Cytarabine	Tirosin-Kinase Inhibitors (Dasatinib)
Ivosidenib, Enasidenib	
Venetoclax	
	Midostaurin

## Data Availability

Not applicable.

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
