# Peer review of "Pericardial Disease in Patients with Cancer: Clinical Insights on Diagnosis and Treatment"

_cancers, 2024, doi:10.3390/cancers16203466_

Round 1
Reviewer 1 Report
Comments and Suggestions for Authors
I think the paper presented is well organized and well presented. This narrative review deals with a disease that often leads to an interruption of oncological therapies and therefore is a topic that has a considerable clinical impact. It is also well written and therefore I believe it is worthy of publication
Author Response
Comment 1: I think the paper presented is well organized and well presented. This narrative review deals with a disease that often leads to an interruption of oncological therapies and therefore is a topic that has a considerable clinical impact. It is also well written and therefore I believe it is worthy of publication
Response 1: Thank you very much for your favorable feedback, we hope this work will help to improve the management of patients with cancer and pericardial effusion.
Reviewer 2 Report
Comments and Suggestions for Authors
- the role of anticoagulation in this setting should be discussed. Authors can consider the paper from Fioretti AM, et al Rev Cardiovasc Med. 2023 Oct 19;24(10):295.
- please include a table gathering the main characteristics of the studies considered from the literateure
Author Response
Comment 1: the role of anticoagulation in this setting should be discussed. Authors can consider the paper from Fioretti AM, et al Rev Cardiovasc Med. 2023 Oct 19;24(10):295.
Response 1:
We deeply appreciate this consideration since it is an important issue in the setting of cancer patients. It is frequent that anticoagulation therapy and pericardial effusion coexist in patients with cancer but there is little evidence. In fact, no recommendations on management of anticoagulant therapy are made in Cardio-Oncology ESC Guidelines (2022). Then, based on our clinical practice, the paper suggested by the reviewer and the recommendations from Pericardial disease ESC Guidelines (2015), a new paragraph explaining this issue has been added (page 9, lines 319 to 325):
“Cancer patients are often on anticoagulation therapy due to cancer-associated thrombosis or atrial fibrillation [60]. Occasionally, pericardial effusion and the need for anticoagulation coexist, leading to a challenging scenario. There are no specific recommendations in the literature about the management of anticoagulation therapy in this setting. In clinical practice, the interruption of the anticoagulant treatment and its rechallenge depend on the evolution, the planned treatment of the pericardial effusion (i.e., percutaneous or surgical treatment) and the indication for anticoagulation [18]”.
Comment 2: please include a table gathering the main characteristics of the studies considered from the literateure
Response 2: Thank you very much for your recommendations following your review. We would like to point out that our review deals with the existing evidence in relation to very different scenarios (acute pericarditis, pericardial effusion and pericardial constriction) of pericardial involvement in cancer patients. The evidence on this pathology is very limited due to the lack of randomized clinical trials (most studies are retrospective and include few patients) and also heterogeneous, making comparison difficult. In this specific case, we did not perform a systematic review/meta-analysis analysing the results of the different studies. For all these reasons, we believe that it would be difficult to produce a comparative table that would be clear to the reader.